# The optical properties of stratospheric aerosol layer perturbation of the Hunga volcano eruption of January 15th, 2022

Pasquale Sellitto[1,2], Redha Belhadji[1], Bernard Legras[3], Aurélien Podglajen[3], Clair Duchamp[3]

[1]Univ Paris Est Creteil and Université de Paris, CNRS, LISA, F-94010 Créteil, France
[2]Istituto Nazionale di Geofisica e Vulcanologia, Osservatorio Etneo, Catania, Italy
[3]Laboratoire de Météorologie Dynamique (LMD-IPSL), CNRS, Sorbonne Université, ENS-PSL, École Polytechnique, Paris, France

*Correspondence to*: Pasquale Sellitto (pasquale.sellitto@lisa.ipsl.fr)

**Abstract.** The Hunga volcano violently erupted on January 15th, 2022, and produced the largest stratospheric aerosol layer perturbation of the last 30 years. In comparison to background conditions and other recent moderate stratospheric eruptions, one notable effect of the Hunga eruption was the significant modification of the size distribution (SD) of the stratospheric aerosol layer, resulting in larger mean particles size and smaller SD spread for Hunga. Starting from satellite-based SD retrievals, and the assumption of pure sulphate aerosol layers, in this work we calculate the optical properties of both background and Hunga-perturbed stratospheric aerosol scenarios using a Mie code. We found that the intensive optical properties of the stratospheric aerosol layer (i.e., single scattering albedo, asymmetry parameter, aerosol extinction per unit mass and the broad-band average UV/Vis-to-MIR Ångström exponent) were not significantly perturbed by the Hunga eruption, with respect to background conditions. The calculated Ångström exponent was found consistent with multi-instrument satellite observations of the same parameter. Thus, the basic impact of the Hunga eruption on the optical properties of the stratospheric aerosol layer was an increase of the stratospheric aerosol extinction (or optical depth), without any modification of the shortwave and longwave relative absorption, angular scattering and broad-band spectral trend of the extinction, with respect to background. This highlights a marked difference of the Hunga perturbation of the stratospheric aerosol layer and those from other larger stratospheric eruptions, like Pinatubo 1991 and El Chichón 1982. With simplified radiative forcing estimations, we show that the Hunga eruption produced an aerosol layer likely 1.5-10 times more effective in producing a net cooling of the climate system with respect to Pinatubo and El Chichón eruptions, due to more effective shortwave scattering. As intensive optical properties are seldom directly measured, e.g. from satellite, our calculations can support the estimation of radiative effects for the Hunga eruption with climate or offline radiative models.

## 1 Introduction

The Hunga volcano (Kingdom of Tonga, 20.54° S, 175.38° W) violently erupted on January 15th, 2022. Due to its very specific shallow submarine volcanological setting, and the subsequent interaction of seawater with the volcanic magma chamber, this eruption was characterised by a large explosivity (e.g. Poli and Shapiro, 2022) and injected volcanic material at altitudes as

large as 56 km, well into the deep stratosphere with parts up to the lower mesosphere (Carr et al., 2022). Basing on a wide array of satellite, ground-based and in situ measurements, it was demonstrated that the Hunga eruption produced the largest perturbation of the global stratospheric aerosol layer since the eruption of Mount Pinatubo (Philippines) in 1991 and the largest perturbation of stratospheric water vapour ever observed (Khaykin et al., 2022, Millán et al., 2022, Sellitto et al., 2022a, Vömel et al., 2022). An uncommonly fast conversion of volcanic sulphur dioxide ($SO_2$) emissions to sulphate aerosols (SA) was observed, with e-folding time from a few days to about 2.0 weeks (e.g. Carn et al., 2022, Asher et al., 2023, Sellitto et al., 2024). This was explained by the gas-to-particle kinetics acceleration due to very large water vapour concentrations, through modelling studies (Zhu et al., 2022). Thus, these very specific perturbations are likely due to the exotic environment for the perturbed stratosphere and drove peculiar chemical and microphysical evolution within the Hunga plume. The Hunga stratospheric aerosol were quickly transported meridionally (weeks-to-months timescales) and stratospheric aerosol perturbations were soon observed encompassing the whole Southern Hemisphere (Legras et al., 2022, Taha et al., 2022). The Hunga aerosol layer showed optical signature of ash only for the very few days after the eruption (Sellitto et al., 2022a). Ash was likely removed from the stratosphere very quickly and its optical signature was not observed after this transient period (Legras et al., 2022, Sellitto et al., 2022a). The Hunga stratospheric aerosol perturbation can then solely be associated to SA. These effects proved long-lasting, with significantly perturbed aerosol extinctions extending well into the years 2022 and 2023 (Duchamp et al., 2023, Sellitto et al., 2024), and probably also in 2024. Using solar-occultation satellite observations, Duchamp et al. (2023) observed aerosol size distributions (SD) in the stratosphere significantly different than usual volcanic perturbations for moderate stratospheric eruptions and the background stratospheric aerosol layer, with significantly larger mean radii and smaller widths of the aerosol size distribution. The liquid-droplets nature and the relatively large mean size of the Hunga aerosol particles was also confirmed with in situ balloon-borne optical counters measurements (Kloss et al., 2022). The aerosol perturbations due to the Hunga eruption are associated with a larger stratospheric aerosol extinction per unit emitted $SO_2$ mass than recent major eruptions, like the one of Pinatubo in 1991, due to the specific aerosol SD in the Hunga plume and the high-altitude $SO_2$ injection (Li et al., 2024). In the absence of volcanic or other perturbations, a local maximum of the vertical aerosol distribution is found in the lower stratosphere, due to troposphere-to-stratosphere exchanges flux of sulphur-containing aerosol precursors at low latitudes and a very limited aerosol sink at these altitudes (Kremser et al., 2016, Norgren et al., 2024). Thus, a background stratospheric aerosol layer can be defined in the absence of episodic perturbation of this layer, i.e. stratospheric volcanic eruptions and pyro-convective fires. Secondary SA largely dominate the composition of the background aerosol layer (Kremser et al., 2016). The stratospheric aerosol layer perturbations induced by the Hunga eruption are expected to have an impact on the optical properties of the stratospheric aerosol layer and on the Earth radiative balance, producing radiative and climatic impacts (Sellitto et al., 2022a).

In this paper, we use the SD determined by Duchamp et al. (2023), for both the Hunga-perturbed and the background stratospheric aerosol layer, and we derive their optical properties. These optical properties and their possible effects on the radiative balance are then compared with those from the major recent eruptions, El Chichón in 1982 and Pinatubo in 1991. Our results are expected to contribute to the new estimations of the radiative forcing of the Hunga eruption, which are presently

ongoing to characterise the long-term radiative impacts of this event. This paper is structured as follows: in Sect. 2 the data and methods used in the paper are described; in Section 3 results are shown and discussed; conclusions are drawn in Section 4.

## 2 Data and Methods

### 2.1 Chemical composition and refractive index

As discussed in the introduction, both the background stratospheric aerosol layer and its perturbation brought by the Hunga eruption can be solely characterised, in terms of composition, as secondary SA particles. Thus, we model both the background and Hunga-perturbed stratospheric aerosol layers as composed of SA. These particles are usually represented as spherical liquid droplets of binary aqueous solution of sulphuric acid ($H_2SO_4$) as done, e.g., by Sellitto and Legras (2016). The link between the chemical composition and the optical properties of a specific aerosol particle is provided by the complex refractive
index (CRI). Among the available laboratory measurements of SA CRI, we have selected, for this work, the one from Hummel et al. (1988). This is one of the few datasets that extend the CRI spectra in both the shortwave (SW) and the longwave (LW) spectral ranges, from the ultraviolet (UV) to part of the far infrared (FIR), for representative stratospheric conditions in terms of the $H_2SO_4$ mass mixing ratio and temperature. Both the background stratospheric SA (e.g. Kremser et al., 2016) and the Hunga perturbations (Duchamp et al., 2023) are characterised by very acidic particles with typically 70-80% $H_2SO_4$ mass
mixing ratio. Thus, a 75% $H_2SO_4$ mixing ratio is selected in this work. Among the available particles temperatures in the Hummel et al. (1988) database, a temperature of 215 K is selected as the more suitable to represent lower and mid-stratospheric conditions. Real and imaginary parts of CRI used in this work are shown in Fig. 1.

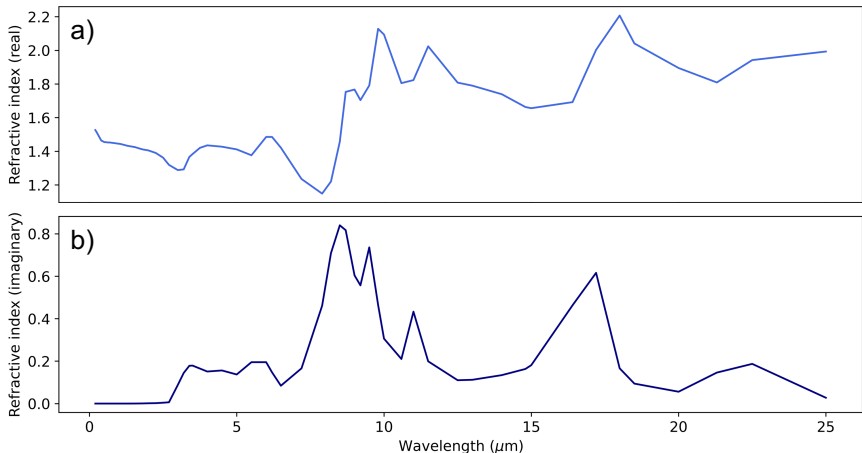

**Figure 1: Real (a) and imaginary part (b) of the complex refractive index of a SA layer with 75% $H_2SO_4$ mass mixing ratio and 215**
**K temperature, from Hummel et al., 1988.**

## 2.2 Size distribution

The number density SD $n(r)$ is defined so that $n(r)dr$ is the number of particles per unit volume, in the aerosol layer, with radius between r and r+dr. All SD, in this work, are modelled as mono-modal log-normal distributions (Eq. 1). In Eq. 1, $N_0$ is the total number concentration (in particles per $cm^{-3}$), $r_m$ is the median radius and $S=\ln\sigma$ is the SD spread i.e. the unitless standard deviation of $\ln(r/r_m)$.

$$n(r) = \frac{N_0}{r\ln\sigma\sqrt{2\pi}} e^{-\frac{1}{2}\left(\frac{\ln\left(\frac{r}{r_m}\right)}{\ln\sigma}\right)^2} \tag{Eq. 1}$$

Both background and Hunga-perturbed typical SD are derived from the results of Duchamp et al. (2023). In that work, SD parameters of a mono-modal log-normal SD, i.e. $N_0$, $r_m$ and $\sigma$ in Eq. 1, are obtained using SAGE III/ISS (Stratospheric Aerosol and Gas Experiment on the International Space Station) satellite observations with the method initially developed by Wrana et al. (2021). Duchamp et al. (2023) applied that method to the Hunga plume and extended back these retrievals to immediately previous unperturbed periods, so to derive SD parameters also for a representative background stratospheric aerosol layer. It is usually convenient, in remote sensing applications, to define an effective radius $r_e$ (the cube particle radius divided by the square particle radius, averaged over the SD); $r_e$ is directly linked to the extinction of the aerosol layer. For a mono-modal log-normal SD, an effective radius can be defined as $r_e = r_m e^{2.5\ln^2\sigma}$. For the background stratospheric aerosol layer a typical combination of $r_m = 0.20$ μm and $\sigma = 1.50$ is considered (Fig. 2a and Tab. 1). The Hunga eruption produced an increase of the mean particle size and decrease of the spread of the particle SD with respect to background conditions (Duchamp et al., 2023). Typical values of Hunga-perturbed stratospheric aerosol layers are used here, with $r_e$ varying between 0.35 and 0.45 μm and $\sigma$ varying between 1.20 and 1.30 (Fig. 2a and Tab. 1). In Fig. 2, $N_0$ is fixed to 1 particle/$cm^3$ for the background and all Hunga cases. As a further comparison, the Mount Pinatubo (Philippines) and El Chichón (Mexico) eruptions in 1991 and 1982 are also considered (Fig. 2b and Tab. 1). For Pinatubo eruption, we use aerosol SD for both a relatively young plume (Asano, 1993) and an extreme case of an aged plume, i.e. several months after the eruption (Russell et al., 1996). Immediately after the Pinatubo eruption, the stratospheric aerosol perturbation was associated with relatively large particles (up to $r_m$=0.6 μm), on average, with an extremely small SD width (down to $\sigma$=1.05) (Asano et al., 1993), The effective radius of the Pinatubo-perturbed aerosol layer increased during the first year after the eruption (extreme values as large as $r_e$=0.9 μm), and then decreased slowly to background levels (Hoffmann and Rosen, 1983). Thus, the SD used in this work for Pinatubo perturbation are to be considered two extreme values for this event. For El Chichón eruption, only a small information is available for the SD and its temporal evolution. In this paper we consider the results of Hoffmann and Rosen (1983), which are representative of the aerosol SD after about 1.5 months after the eruption of El Chichón.

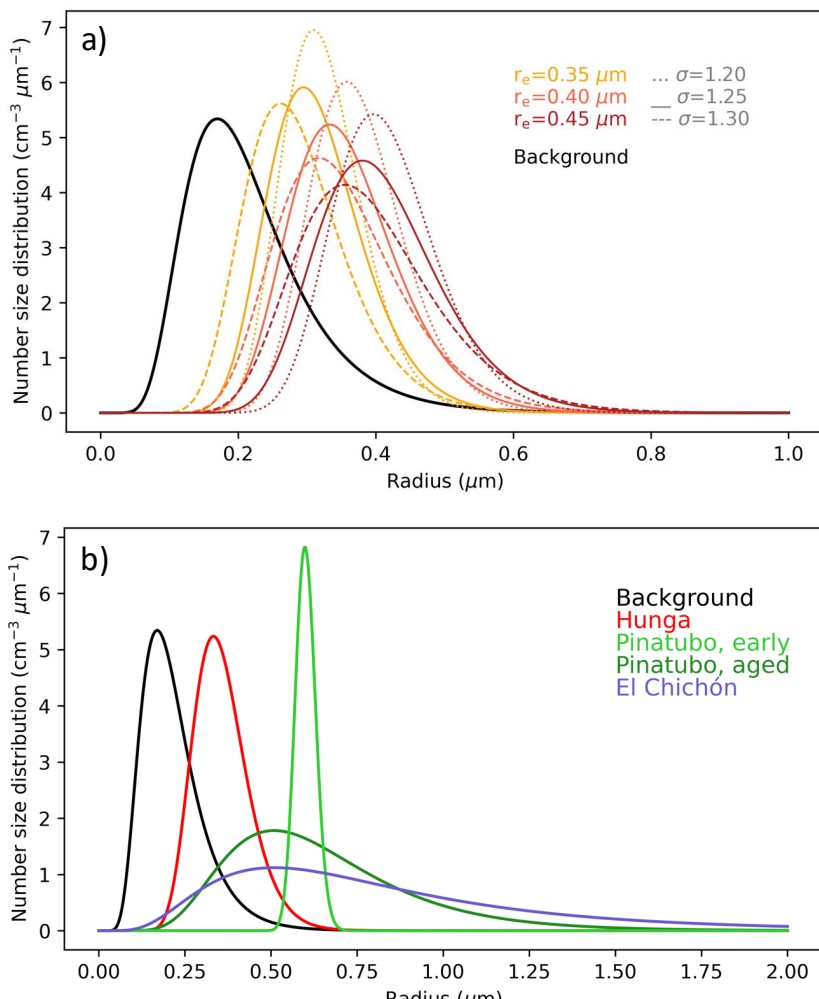

**Figure 2: Typical background (black curve) and Hunga-perturbed SDs (yellow, red and dark red curves), modelled as mono-modal log-normal SDs with $r_e$ and $\sigma$ estimated using SAGE III/ISS by Duchamp et al. (2023) (a). Background (black curve), Hunga-perturbed (red curve, case for $r_e = 0.40$ μm and $\sigma = 1.25$), Mount Pinatubo-perturbed (2 cases: early plume, light green curve, and aged plume, dark green curve) and El Chichón-perturbed SD (blue curve) (b). Log-normal parameters for Pinatubo (young and aged plume) and El Chichón SD are taken from Asano, 1993, Russell et al., 1996 and Hoffmann and Rosen, 1983, respectively. See Tab. 1 for more details on the SD parameters of all cases shown in the figure. Please note the different x-axis intervals in panels a and b.**

For a SA layer of given mono-modal log-normal SD, the effective SA mass $M_e$ (i.e. the total aerosol mass per unit volume in the layer) can be calculated as done in Eq. 2, where ρ is the mass density of SA (here taken as 1.75 g/cm³) and $N_e = N_0 e^{-3ln^2\sigma}$ is the mono-modal log-normal effective number density. The $M_e$ values for the SDs used in the present work are also indicated in Tab. 1.

$$M_e = \frac{4}{3}\pi r_e^3 \rho N_e \qquad \text{(Eq. 2)}$$

**Table 1: Parameters $r_m$, $r_e$, $\sigma$ and $M_e$ for the SD in Fig. 2a for the background and Hunga-perturbed stratospheric aerosol layers used in this manuscript, as well as Mount Pinatubo-perturbed (2 cases: early plume and aged plume) and El Chichón-perturbed SD (blue curve). All SD in Fig. 2a are for a value of $N_0$=1 particles/cm³. In the table, $N_0$ values for each SD are also reported, in the case of a fixed $M_e$=1 $\mu$g/m³ (last column).**

| $r_m$ ($\mu$m) | $r_e$ ($\mu$m) | $\sigma$ | $M_e$ ($\mu$g/m³) | $N_0$ (#/cm³) |
|---|---|---|---|---|
| **Background** | | | | |
| 0.20 | 0.30 | 1.50 | 0.12 | 8.27 |
| **Hunga-perturbed** | | | | |
| 0.32 | 0.35 | 1.20 | 0.28 | 3.52 |
| 0.31 | 0.35 | 1.25 | 0.27 | 3.69 |
| 0.28 | 0.35 | 1.30 | 0.26 | 3.91 |
| 0.37 | 0.40 | 1.20 | 0.42 | 2.36 |
| 0.35 | 0.40 | 1.25 | 0.40 | 2.47 |
| 0.34 | 0.40 | 1.30 | 0.38 | 2.62 |
| 0.41 | 0.45 | 1.20 | 0.60 | 1.65 |
| 0.40 | 0.45 | 1.25 | 0.55 | 1.74 |
| 0.38 | 0.45 | 1.30 | 0.54 | 1.84 |
| **Pinatubo-perturbed, young plume (Asano, 1993)** | | | | |
| 0.60 | 0.60 | 1.05 | - | 0.64 |
| **Pinatubo-perturbed, aged plume (Russell et al., 1996)** | | | | |
| 0.60 | 0.90 | 1.50 | | 0.31 |
| **El Chichón-perturbed (Hoffmann and Rosen, 1983)** | | | | |
| 0.72 | 1.71 | 1.80 | - | 0.08 |

## 2.3 Optical properties calculation

The optical properties of background, and Hunga-, El Chichón- and Pinatubo-perturbed stratospheric aerosol layers are calculated with the scheme shown in Fig. 3. Starting from CRI and SD input data described in Sects. 2.1-2, a Mie code estimates the extinction coefficient ($\beta_{ext}$), single scattering albedo (SSA) and asymmetry parameter (g) spectra for both cases, and for Mount Pinatubo (young and aged plume) and El Chichón as a further comparison. As Mie code, the IDL (Interactive Data Language, http://www.exelisvis.com/ProductsServices/IDL.aspx) Mie scattering routines of the Earth Observation Data Group of the Department of Physics of Oxford University are used (https://eodg.atm.ox.ac.uk/MIE/). The outputs of the Mie calculations are intended to represent the overall extinction of typical stratospheric background and volcanically-perturbed layers, as well as their absorption/scattering properties (through SSA) and angular distribution of the scattered radiation (through g). The SSA is the ratio of the scattering to extinction (i.e. scattering plus absorption) efficiency factors $Q_{sca}$ and $Q_{ext}$ obtained with Mie calculations (van de Hulst, 1957). Values of the SSA approaching 1.0 point at pure scattering particles, while values of the SSA approaching 0.0 point at pure absorbing particles. The g parameter is the mean value of the cosine of the scattering angle weighted though the scattering phase function obtained with Mie calculations (van de Hulst, 1957). Values of the g parameter approaching 1.0 point at pure forward scattering, while values of the g parameter approaching 0.0 may point at isotropic scattering. The overall extinction of the layers, measured by $\beta_{ext}$, has been calculated with two different assumptions: a fixed $N_0$ and a fixed $M_e$. With this latter, extinction spectra per unit mass concentration ($\beta_{ext}/M_e$) have also been derived. The SSA, g and $\beta_{ext}/M_e$ are intensive optical properties of the aerosol layer, i.e. they do not depend on the injected aerosol mass. It was discussed previously that these optical parameters, together with surface reflectivity information, fed to a radiative transfer model (RTM), are sufficient to describe the radiative properties and impacts of an aerosol layer, like the instantaneous radiative forcing and the vertical profiles of radiative/cooling heating rates (e.g. Sellitto et al., 2022b, 2023). In this work, optical properties in both the solar shortwave (SW) and the terrestrial longwave (LW) spectral ranges are estimated, so to produce results usable in RTM when interested in the whole radiant energy spectrum in Earth's atmosphere. The output spectral range of the optical properties is somewhat limited by the available spectral range of the input CRI. In our case, we are limited at 25 μm on the upper end, due to available SA CRIs in the literature, so a part of the far infrared range is not represented in this work likely leading to an underestimation of its impact.

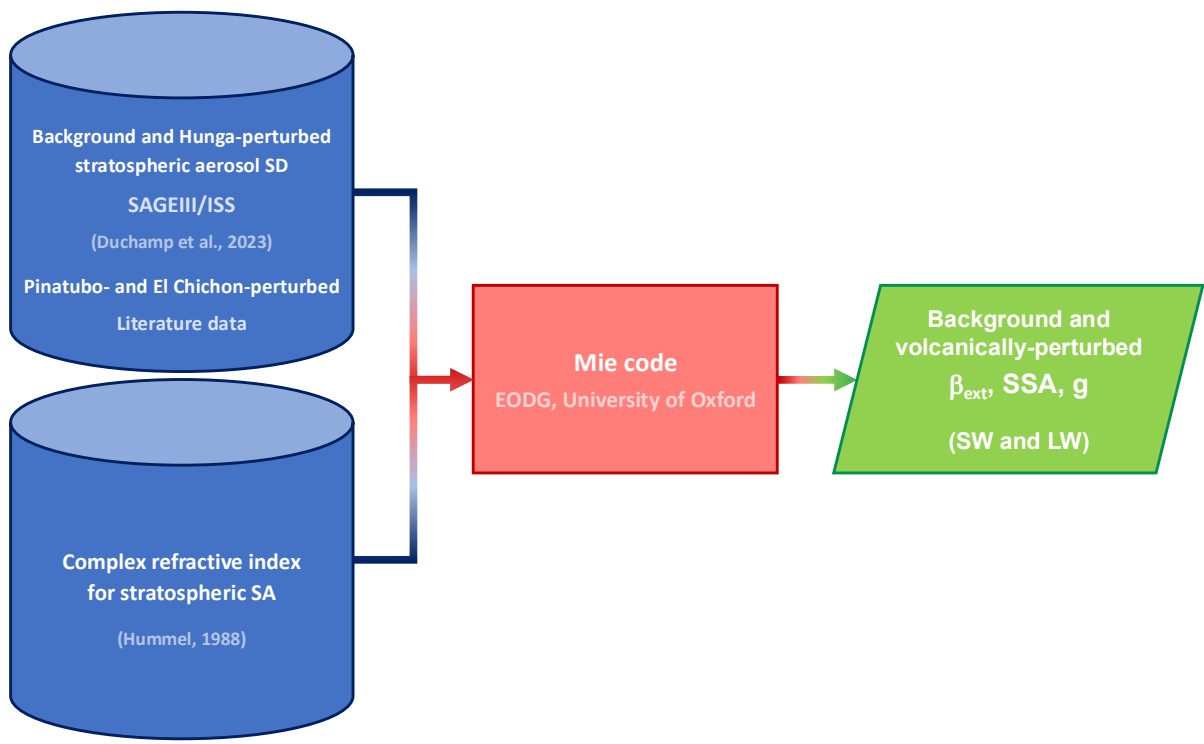

**Figure 3: Schematic of the optical properties calculations used in this work.**

## 3 Results and discussion

### 3.1 Impact of size distribution on the optical properties

Figure 4 shows $\beta_{ext}/N_0$ and $\beta_{ext}/M_e$ calculation, for the background and Hunga-perturbed scenarios, as described in Sect. 2. The extinction is an extensive optical property, which depends on the aerosol layer mass (the SA mass, in the present case). Thus, large differences in $\beta_{ext}/N_0$ among background and Hunga-perturbed scenarios can be observed (Fig. 4a). This is principally due to the larger SA mass, for a fixed number density $N_0=1.0$ #/cm$^3$, of the larger Hunga particles with respect to the smaller background particles. The effective mass varies from 0.12 μg/cm$^3$, for the background layer, to up to 0.60 μg/cm$^3$, for the Hunga-perturbed layers (see Tab. 1). On the contrary, $\beta_{ext}/M_e$ is much less variable depending on the scenario, except at very short wavelengths in the SW (Fig. 4b). In general, a clear exponential decrease of $\beta_{ext}$ can be observed in the UV/Vis and part of the NIR (thus, in most of the SW spectral range), following the empirical Ångström law in Eq. 3, where λ is the wavelength ($\lambda_{ref}$ is a reference wavelength, in many cases taken as 1 μm) and AE is the Ångström exponent.

$$\beta_{ext}(\lambda) = \beta_{ext}(\lambda_{ref})\lambda^{-AE} \tag{Eq. 3}$$

At longer wavelengths, absorption features of SA particles appear, including the peculiar MIR signatures at 8.0 to 11.0 μm. These absorption features can be associated with the rotational-vibrational absorption bands of the undissociated $H_2SO_4$ in the concentrated solution droplets discussed e.g. by Sellitto and Legras (2016). More absorption features are visible in the NIR, from about 3.0 to 6.0 μm, and in the FIR, from 15.0 to 18.0 μm. The two regimes, dominated by scattering in the SW, for wavelengths shorter than about 3.0 μm, and by absorption in the LW, for longer wavelengths, is discussed further in the following (associated with SSA), and is linked to a large variability of $\beta_{ext}/M_e$. This latter ranges between values up to 4.5 $10^{-3}$ km$^{-1}$ per unit effective mass, below 3.0 μm, to less than 1.0 $10^{-3}$ km$^{-1}$ per unit effective mass, above 3.0 μm.

Figure 5 shows the SSA (Fig. 5a) and g (Fig. 5b) calculation, for the background and Hunga-perturbed scenarios. One first take-away message from these results is that both SSA and g are not perturbed significantly by Hunga eruption, and then their absolute values and variability are similar for background and Hunga-perturbed scenarios. The two scattering- and absorption-dominated regimes, discussed for the aerosol extinction, can be seen here with SSA approximately 1.0 (pure scattering aerosol layers), for wavelength shorter than about 3.0 μm, and sharply decreasing to values lower than about 0.2, for wavelength longer than about 3.0 μm. The SSA approaches values of 0.0 (pure absorbing particles), for wavelengths longer than about 6.0 μm. The g parameter, starting from values of about 0.6 to 0.8, steeply decreases to values lower than 0.1, for wavelengths longer than 6.0 μm. This can be associated with a markedly dominating forward scattering in the SW and a quasi-isotropic scattering in the LW. Figures 4 and 5 are also shown on a vertical log-scale in Fig. S1.

These results are summarized in Fig. 6, where spectral-band average values of the intensive optical properties $\beta_{ext}/M_e$, SSA and g, for background and Hunga-perturbed layers, are shown for the UV/Vis, NIR, MIR and FIR broad-band ranges. Large values for the three intensive parameters are found in the UV/Vis, thus pointing at very extinction-effective layers, dominated by a largely forward scattering. While this characteristic behaviour of the aerosol layers stands for the whole SW, the extinction-efficiency markedly decreases in the NIR. In the LW, the optical characterisation of the layers is more spectrally-homogeneous, with relatively small extinction dominated by absorption and a small quasi-isotropic scattering, with small differences between the MIR and the FIR.

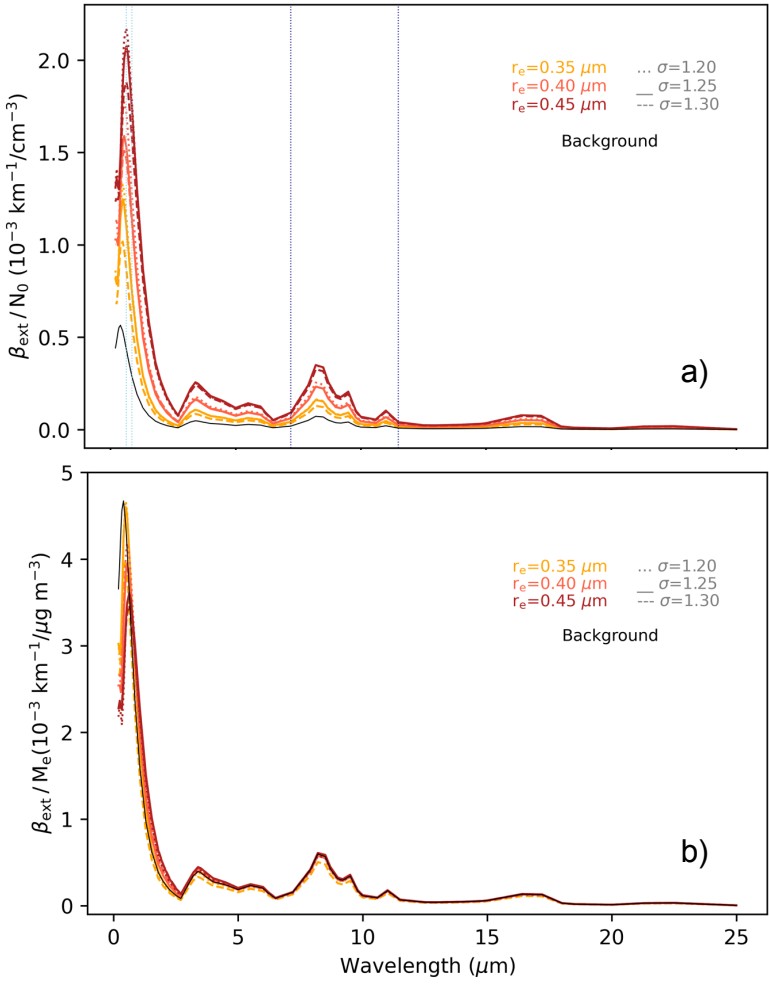

**Figure 4: Extinction coefficient spectra $\beta_{ext}$ at fixed number concentration $N_0$ (a) and effective mass $M_e$ (b), for background (black curve) and Hunga-perturbed aerosol layers (from yellow, red and dark red dotted, dashed and solid curves – see corresponding values of the $r_e$ and $\sigma$ assumption in the figures captions). Light and dark blue vertical dotted lines in panel a indicate the two wavelength bands (UV/Vis and MIR, respectively) used to average the extinction to calculate the UV/Vis-to-MIR AE of Fig. 7.**

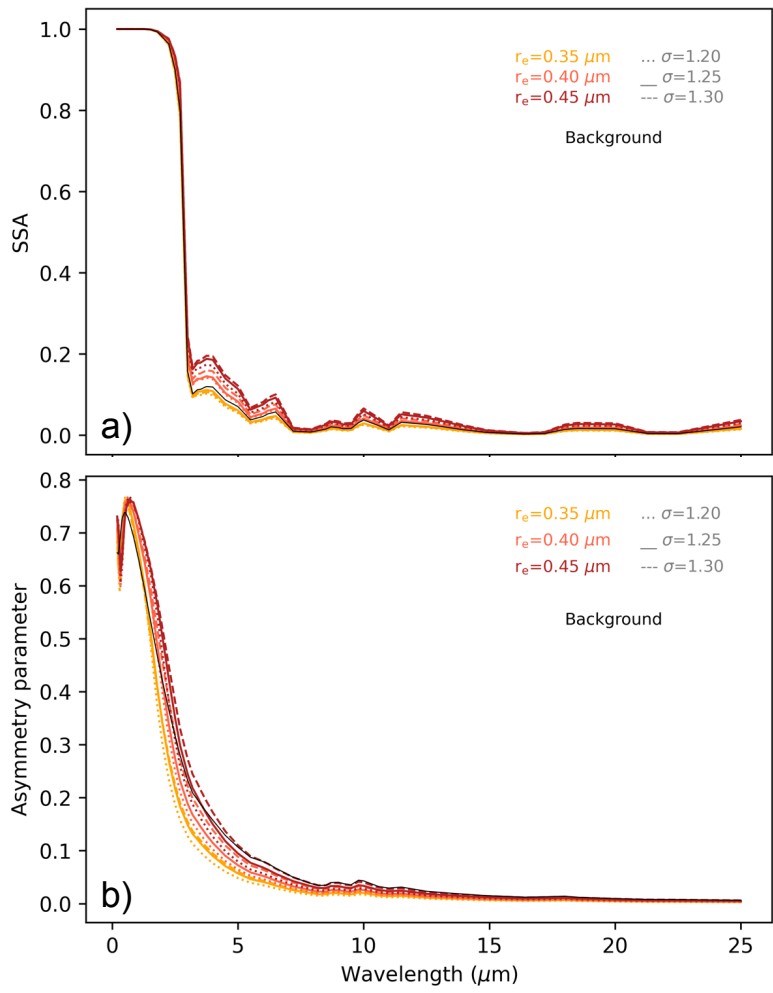

**Figure 5: Same as Fig. 4 but for single scattering albedo SSA (a) and asymmetry parameter g (b) spectra.**

More in general, the most apparent effect, visible in Figs. 4-6, is that all intensive optical properties of Hunga-perturbed layer are very similar to those in background conditions. It can be concluded that, despite the significant perturbation in stratospheric aerosol SD and the large SA injected mass, the Hunga eruption had minimal effects on the intensive optical properties of the stratospheric aerosol layer. Thus, the only impact of Hunga eruption on the optical properties of the stratospheric aerosol layer is the marked increase in the overall extinction and the aerosol optical depth (AOD). For this latter property, Hunga produced the largest global perturbation since Mount Pinatubo eruption in 1991 (Sellitto et al., 2022a). The specific absorption and scattering properties of the stratospheric aerosol layer were not significantly perturbed in the SW and LW by the Hunga eruption, as shown by SSA and g calculations of Fig. 5 and 6b,c. It must be noted that Hunga perturbations on the stratospheric aerosol SD, though significant, are much smaller than for other stronger recent eruptions like El Chichón in 1982 (Hofmann and Rosen, 1983) or the aged Pinatubo in 1991 (e.g., Russell et al., 1996), see respective SDs in Fig. 2b. In these cases, a

fraction of relatively large particles (larger than 1.0 μm) was present, which is not the case for the Hunga eruption (see Fig. 2b). For the fresh Pinatubo plume, i.e. during the first few months after the eruption in 1991, larger particles were present, on average, than for the Hunga eruption, but with a minimal contribution of particles larger than 1.0 μm, due to the exceptionally

small width of its SD (Asano, 1993). We performed additional Mie calculations for Pinatubo and El Chichón SDs, using the SD parameters of Asano (1993), Russell et al. (1996) and Hoffmann and Rosen (1983), summarised in Tab. 1. For the aged Pinatubo and El Chichón cases, the UV/Vis $\beta_{ext}/M_e$ is significantly smaller, and the overall LW SSA and g are significantly larger, than for Hunga and stratospheric background (Fig. 6). This effect, even if still present, is less strong for the fresh Pinatubo plume. Such a change of optical regime implies a stronger LW climate warning effect, that can counterbalance the

SW cooling. More in general, Lacis et al. (1992) have shown that a marked change in optical and radiative regime occurs for volcanic aerosol layers with mean size exceeding 1.0 μm, which is the case for the aged Pinatubo plume and El Chichón but not for the Hunga perturbations. This change in the SW/LW optical properties can result in less effective cooling at top of the atmosphere or even in aerosol-related warming, which could have been initially the case for Pinatubo and El Chichón eruptions (before a marked and relatively long-term cooling effect, due to the removal of larger aerosol particles) but was likely not the

case for Hunga, even in the first phases after the eruption (e.g. Sellitto et al., 2022a). The mean particle size for Hunga was larger than any post-Pinatubo stratospheric eruption (Wrana et al., 2023). Most post-Pinatubo eruptions perturbed the stratospheric aerosol layer with a decrease of mean particle size, rather than an increase.  The radiative effects of the Hunga-related aerosols SD perturbations are discussed more in Sect. 3.2.

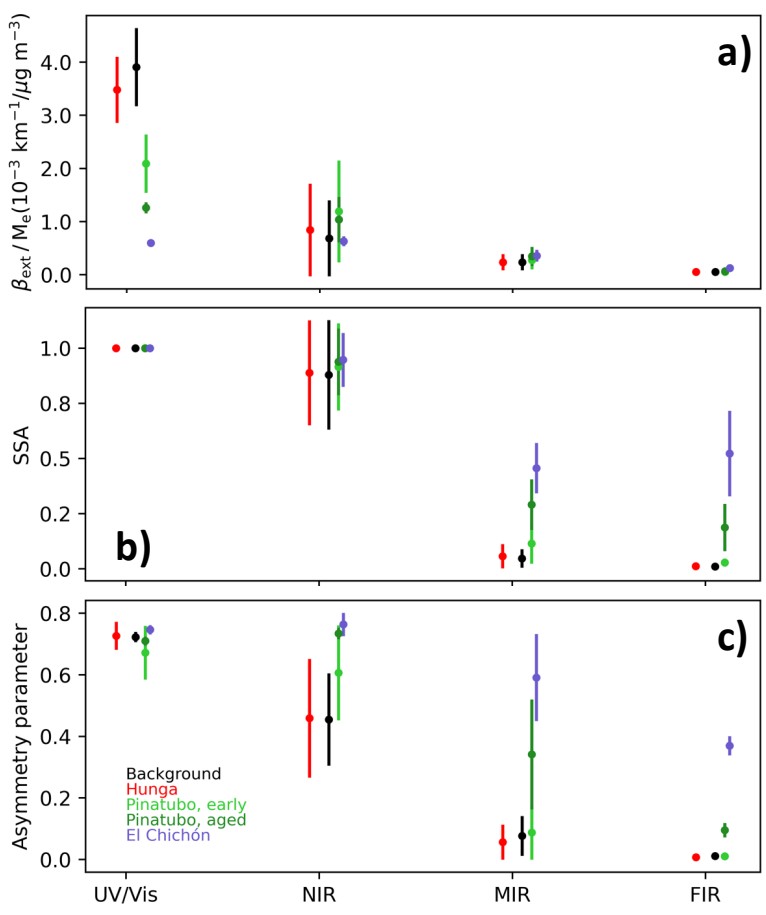

**Figure 6: Band average (UV/Vis, NIR, MIR and FIR) extinction coefficient per unit effective mass (a), single scattering albedo (b) and asymmetry parameter (c), for background (black circles with error bars) and Hunga-perturbed (averaged over all scenarios in e.g. Fig. 2, red circles with error bars) stratospheric aerosol layer. Band-average values of these intensive aerosol optical properties are also shown for Pinatubo (2 cases: early plume, light green circles with error bars, and aged plume, dark green circles with error**
**bars) and El Chichón (blue circles with error bars). UV/Vis (ultra-violet and visible): 0.30-0.85 mm; NIR (near infrared): 0.85-3.0 mm; MIR (mid infrared): 3.0-15.0 mm; FIR (far infrared): 15.0-25.0 mm.**

Another intensive aerosol optical parameter is the Ångström exponent AE, see also Eq. 3. An average AE can be obtained by combining aerosol extinction or AOD information at two different wavelengths. In Sellitto et al. (2024) the average UV/Vis-to-MIR AE was calculated using combination of OMPS-LP (Ozone Mapping and Profiler Suite – Limb Profiler) and IASI

(Infrared Atmospheric Sounding Interferometer) observations, at 0.7 and 8.5 μm, respectively. As for other intensive optical properties shown above, the stratospheric aerosol layer UV/Vis-to-MIR AE obtained using our calculations is not significantly perturbed by the Hunga eruption, and background and Hunga-perturbed values stay around a value of 1.0 (0.94, for the background stratospheric conditions, and 0.97±0.02, for Hunga-perturbed conditions, with standard deviation representing the variability associated with the different Hunga SD). These results are not dissimilar from the observed value (1.13±0.23), even

if ~15% smaller. Thus, this shows the consistency of our calculated and the observed optical properties for the Hunga plume.

It has to be noted that Taha et al. (2023) have shown that the observed Vis AE, i.e. estimated with two different Vis bands of the OMPS-LP instrument (0.52 and 1.02 µm), is significantly perturbed by the Hunga eruption. The AE estimation from observations is very sensitive to the selection of the two wavelengths used to derive it. Using the same wavelengths as in Taha et al. (2023), we obtain Vis AE of 1.35 and 0.80 for the background and Hunga-perturbed scenarios. This is very consistent with past theoretical studies, i.e. the one of Schuster et al. (2006) (see their figure 4a, with effective radii 0.30 µm and 0.40-0.45 µm, respectively). The variability of the UV/Vis AE, in terms of the selected background or Hunga-perturbed scenarios, can also be seen Fig. 4, reflected by the different slopes of the spectral variability of the extinction coefficient.

## 3.2 Impact of optical properties on the radiative forcing

A simple parameterisation of the top-of-atmosphere (TOA) radiative forcing is used to estimate the radiative impact, through optical properties perturbations, of changing SD of stratospheric aerosol by Hunga eruption. For partly absorbing aerosols in a layer over a given surface of spectral reflectivity $R_s$ and placed in a stratified atmosphere of spectral transmissivity $T_{atm}$, a broad-band TOA radiative forcing $\Delta F$ per unit SA mass can be defined as in Eq. 4. This quantity is estimated at the four broad-band spectral ranges, as defined, e.g., in Fig. 6 (UV/Vis, NIR, MIR and FIR) and then these broad-band estimations are added up to obtain a total $\Delta F$. In Eq. 4, the optical properties $\beta_{ext}/M_e$ and SSA are the same as in Fig. 6. The optical parameter b is the hemispheric backscatter ratio, which can be derived using our asymmetry parameter g estimations, using the method described by Marshall et al. (1995). The aerosols have been put in a 1-km deep layer. In the parameterisation, S is the radiation source, taken as pure Planck functions at temperatures 5770 K, in the SW range (UV/Vis and NIR broad bands) and at the temperature 300 K, in the LW range (MIR and FIR broad bands), to simulate the solar and the terrestrial radiation sources. The solar S component is then scaled at the mean sun-Earth distance. The diurnal cycle of S was not considered in this example. The underlying surface have been considered as marine ($R_s$ 0.05 in the UV/Vis and FIR and 0.00 in the NIR and MIR) and average values of $T_{atm}$ have also been considered (0.6, 0.5, 0.5 and 0.2 in the UV/Vis, NIR, MIR and FIR).

$$\Delta F = -S(\lambda)T_{atm}(\lambda)\frac{\beta_{ext}(\lambda)}{M_e}SSA(\lambda)b(\lambda)\left(\left(1-R_s(\lambda)\right)^2 + 2R_s(\lambda)\left(\frac{1-SSA(\lambda)}{b(\lambda)SSA(\lambda)}\right)\right)\Delta\lambda \qquad \text{(Eq. 4)}$$

The SW (UV/Vis + NIR), LW (MIR + FIR) and total (SW+LW) radiative forcings obtained with Eq. 4 and our estimated optical parameters are shown in Fig. 7, for a background stratospheric aerosol layer, and layers perturbed by Hunga, Pinatubo and El Chichón eruptions. Two different radiative regimes can be observed in Fig. 7. The first regime, for smaller effective radii, is clearly dominated by the SW scattering, thus resulting in a large negative radiative forcing per unit mass. The second regime, for larger effective radii, is less and less dominated by the decreasing SW scattering with increasing mean size and the progressively increasing importance of the LW warming effect due to LW absorbing effect. In this latter case, the total radiative forcing per unit SA mass decreases significantly with respect to background conditions. In addition, SW/LW absolute ratio of the $\Delta F$ is approximately 99, 98, 80 and 60% for the background, Hunga, aged Pinatubo and El Chichón scenarios. These results

point at a very different overall radiative regimes between background and Hunga, with respect to aged Pinatubo and El Chichón. For the young Pinatubo plume, the SW/LW absolute ratio of the $\Delta F$ is around 90%, larger than for the aged Pinatubo plume, while still smaller than for the Hunga-perturbed scenario. The Hunga eruption did not modify significantly the radiative regime of the stratospheric aerosol layer with respect to background conditions, with values around -0.17 W/m$^2$ per µg/m$^3$ SA, dominated by SW scattering (pink triangle in Fig. 7). This is 1.5 to 4 times more effective, per unit SA mass, than Pinatubo eruption (about -0.12 to -0.040 W/m$^2$ per µg/m$^3$ SA, depending on the ageing of the plume, pink squares) and nearly 10 times more effective than El Chichón eruption (about -0.015 W/m$^2$ per µg/m$^3$ SA, pink hexagon). For these latter, the SW cooling markedly decreased, as well as the SW/LW ratio, with respect to background and Hunga eruption. Our results are consistent with previous studies, in particular the one of Lacis et al. (1992) who showed that the cooling potential at TOA of stratospheric aerosols significantly decreases for effective radii larger than about 1.0 µm, and can even switch from negative (cooling) to positive (warming) for effective radii larger than 2.0 µm (see their Fig. 2). The effect of a significantly larger RF per unit SA mass, with respect to major recent eruptions, adds up with the larger stratospheric AOD production per unit SO$_2$ injected mass shown by Li et al. (2024). Thus, for different factors, including the high-altitude injection and the aerosol SD, the Hunga eruption produced stratospheric aerosol perturbations with a particularly large potential to produce a negative RF and a cooling of the climate system. These factors can be associated with the phreatic nature of the eruption and, so, the interaction with seawater before the eruption and the subsequent availability of water vapour in the plume. It is important to recall that Hunga eruption is characterised by much less injected SO$_2$ and SA mass than Pinatubo and El Chichón eruptions so, despite the larger radiative effectiveness of its stratospheric aerosol perturbations, in terms of the SD, its overall radiative effect is expected to be smaller than Pinatubo and El Chichón. The direct radiative effect of water vapour must also be considered. Water vapour can produce a warming at TOA, that can bias or revert the cooling effect of the sulphate aerosol layer (Sellitto et al., 2022a).

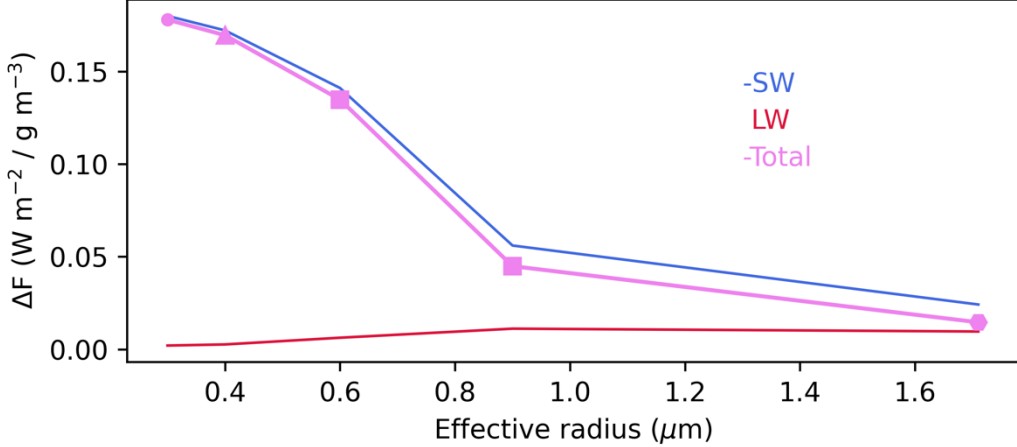

**Figure 7: TOA RF (ΔF) in the SW (negative, dark blue lines), LW (dark red lines) and total SW+LW (negative, pink lines and symbols) as a function of the effective radius for: background conditions (pink dot), Hunga (pink triangle), and Pinatubo (pink squares) and El Chichón (pink hexagon), see text for details.**

Optical properties of aerosol layers are needed, in radiative calculations with both climate models and offline radiative models, for a given source of radiative forcing to estimate its impacts (e.g. Sellitto et al., 2022b, 2023). Our results provide a ready-to-use benchmark for the radiative impacts estimations of the Hunga eruption versus stratospheric aerosol background. Thus, we provide as Supplementary datasets optical properties: (1) computed at the wavelengths at which the refractive indices of Hummel et al. are available, and (2) averaged over the 30 broad bands of the Rapid Radiative Transfer Model (RRTM, Iacono

et al., 2008) in the ECMWF ECRAD implementation (Hogan, and Bozzo, 2018) (these latter are provided as a netCDF file), for future studies on the Hunga radiative impacts. Averaged values for the UV/Vis, NIR, MIR and FIR bands are also reported in Tab. 2 (same as Fig. 6). Caution must be taken for the radiative impacts estimation of Hunga eruption because of the additional important radiative impacts of water vapour injections, that can be even larger than aerosol impacts during the first months (Sellitto et al., 2022). This additional radiative effect for the Hunga eruption was not present in recent stratospheric

volcanic eruptions from subaerial volcanoes.

**Table 2: Summary of band-average ready-to-use optical properties inputs for radiative calculations**

| Spectral interval (µm) | Spectral interval acronym | AE Background | SSA Background | g Background | AE Hunga-perturbed | SSA Hunga-perturbed | g Hunga-perturbed |
|---|---|---|---|---|---|---|---|
| 0.3-0.8 | UV/Vis | | 0.99±0.00 | 0.72±0.02 | | 0.99±0.00 | 0.73±0.05 |
| 0.8-3.0 | NIR | 0.94 | 0.88±0.25 | 0.45±0.15 | 0.97±0.02 | 0.89±0.24 | 0.46±0.19 |
| 3.0-15.0 | MIR | | 0.05±0.04 | 0.08±0.06 | | 0.06±0.06 | 0.06±0.06 |
| 15.0-25.0 | TIR | | 0.01±0.01 | 0.01±0.00 | | 0.01±0.01 | 0.01±0.00 |

## 4 Conclusions

The submarine Hunga volcano violently erupted on January 15[th], 2022 and produced the largest stratospheric aerosol layer perturbation, in terms of the aerosol extinction and the injected aerosol mass, since the climate-relevant eruption of Pinatubo volcano in 1991. One notable feature of this perturbation is the significant modification of the stratospheric aerosol SD, with larger particles (0.35-0.45 µm effective radius) and a smaller SD width (1.20-1.30 spread in a modelled mono-modal size distribution) than the background stratospheric aerosol layer and than for most other post-Pinatubo stratospheric eruptions. In this paper, using a Mie code and the assumption of pure SA layers, we calculate the optical properties of the Hunga-perturbed stratospheric aerosol layer and compare with those for a background scenario. We found that, despite the sensible impact on the aerosol SD, the Hunga eruption had only a minor impact on the intensive optical properties, namely the aerosol extinction per unit effective mass $\beta_{ext}/M_e$, the single scattering albedo SSA, the asymmetry parameter g and the broad-band UV/Vis-to-MIR Ångström exponent AE. Thus, while producing an historical perturbation on the stratospheric aerosol extinction and optical depth, the Hunga eruption did not modify the absorption and angular-scattering properties of the stratospheric aerosol layer. Our UV/Vis-to-MIR AE calculations are consistent with those observed with the combination of OMPS-LP and IASI satellite observations. We further calculate intensive aerosol optical properties for past eruptions of Pinatubo 1991 and El Chichón 1982 and found that these latter events produced enough particles with radii > 1 µm to change their SW/LW optical regimes with respect to stratospheric background, which was not the case for Hunga. We demonstrate here, with a simplified radiative forcing parameterisation, that the Hunga eruption has likely produced stratospheric aerosol layers with a 1.5 to 10 times larger cooling potential than the Pinatubo and El Chichón eruptions, due to SD-related different radiative regimes.

Hunga-related stratospheric aerosol perturbations are more effective in SW cooling than Pinatubo- and El Chichón-related perturbations. Values as large as -0.17 W/m$^2$ per μg/m$^3$ of injected SA are found for Hunga eruption (as comparisons: for Pinatubo, values of -0.012 to -0.040 W/m$^2$ per μg/m$^3$ of injected SA are found; for El Chichón, values as small as -0.015 W/m$^2$ per μg/m$^3$ of injected SA are found). Our study highlight the importance of having detailed information on the aerosol SD to obtain reliable estimations of their radiative impacts. Our optical properties calculations are a ready-to-use dataset for future estimations of the radiative impacts of the Hunga eruption compared with background, combined with pertinent water vapour observations, which is another important forcing agent for this submarine eruption.

## Author contribution

PS conceived and performed the study and wrote the first version of the manuscript. All authors participated to the discussions of the results and contributed to the final version of the manuscript.

## Competing interests

At least one of the (co-)authors is a member of the editorial board of Atmospheric Chemistry and Physics. The peer-review process will be guided by an independent editor. The authors also have no other competing interests to declare.

## Acknowledgements

The optical properties used in this work are obtained with the IDL Mie scattering routines developed by the Earth Observation Data Group of the Department of Physics of Oxford University, and available via the following website: http://eodg.atm.ox.ac.uk/MIE/.

## Financial support

This research has been supported by the Centre National d'Etudes Spatiales (grant: EXTRA-SAT).

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
