# Peer review of "The optical properties of stratospheric aerosol layer perturbation of the Hunga volcano eruption of January 15th, 2022"

_EGUsphere, 2024_

## Author Comment (AC1)

**Review of the manuscript "The optical properties of stratospheric aerosol layer perturbation of the Hunga volcano eruption of January 15th, 2022"**

Dear Editor, dear anonymous Reviewers #1-2,

Many thanks for your constructive criticism and the very useful comments. Thanks to your commitment to the review process, we have thoroughly revised our manuscript. Please find more details and a point-by-point reply to the Reviewers' specific comments in the following (Reviewers' comments are in black and our replies in blue). We think that, thanks to your comments and suggestions, the present version of the manuscript is greatly improved with respect to the previous version.

Thank you very much,
Pasquale Sellitto on behalf of all co-authors

**Reviewer #1**

Sellitto et al. built on published results of the Hunga aerosol size distribution to derive the optical properties characteristic of the volcanic plume. They present original results and compare them to historical volcanic events, El Chichon and Mount Pinatubo eruptions and their subsequent plume properties. The authors use a suite of instrument and publicly available datasets to constrain a Mie code allowing a closure on the optical, microphysical and radiative properties of these sulfate aerosols. The manuscript is generally well written and organized. Given its magnitude and particularities, the Hunga eruption has resulted with a strong interest from the Earth Science community, in particular atmospheric scientists. Numerous studies have already been published aiming at characterizing the Hunga aftermath, and this study is adding to this corpus with a new angle. On the downside though, I found a lack of references to acknowledge the work already achieved by peers. Although it presents some new results, a better referencing of previous studies may underline even more why this work is impactful. After consideration of minor revisions, this work should be suitable for publication in ACP.

Thank you very much for your kind words and your constructive criticism. Please find a point-by-point reply to your main (MaC) and minor comments (MiC) in the following.

**Main comments:**

MaC1) Introduction in general: it is well written but could use more references to highlight what is already known and publish and where this study sits in that context. Li et al. (2024) (https://doi.org/10.1029/2024GL108522), for example, presents some similar computation on the extinction efficiency of the Hunga aerosol compared to Pinatubo.

We cited now Li et al. (2024), which is also added to the discussion, later in the text. Please note that Li et al. Calculate the aerosol extinction efficiency as a step to estimate the stratospheric AOD per unit SO2 emitted mass. This is complementary but not at all overlapping with our calculations and considerations – the fact that Hunga's plume optical properties produce a more effective negative radiative forcing than for other recent eruptions. The two evidences, our and those from Li et al., actually cumulate towards a more effective radiative forcing: 1) Hunga produces a plume with more stratospheric AOD per unit injected SO2 mass (Li et al.), and 2) specific intensive optical properties for this eruption, produce more radiative effective plume (this work). We discuss this in the manuscript.

MaC2) In section 2.3, it is stated that Mie calculations are performed in two ways with two assumptions Me fixed or N0 fixed. These two quantities are linked together through Eq.2. The authors should clarify what is the benefit of doing the computations as such.

This is actually explined in the following line ("The SSA, g and $b_{ext}/M_e$ are intensive optical properties of the aerosol layer, i.e. they do not depend on the injected aerosol mass.").

MaC3) Figures 4 and 5, the linear scale renders the readability difficult as all the curves are close together. Enhancing the figures with some zooms or a logscale may help. Additionally, although the IR and FIR are crucial for radiative budget, a lot of the variability occurs in the SW UV/Vis, I would also suggest a finer resolution in this part of the spectrum. In particular, we see an increase in the extinction efficiency from the UV to Vis, is this expected? Even for the background state?

The spectral resolution of Mie calculations is limited by the spectral resolution of the available complex refractive indices used in input (in this case, the ones from Hummel et al.), which is relatively coarse. Thus, we cannot do a better work in terms of spectral resolution in Figs. 4 and 5. Most of the interest in our study comes to the variability of optical properties (and then radiative properties) from SW to LW: this is much better shown in a linear scale than a log scale. By the way, we now provide additional figures in log scale in the Supplementary Material.

MaC4) It looks like the authors already have the necessary parameters to plot the El Chichon and Pinatubo cases alongside the background and the different Hunga cases of Figure 4 and 5. It would be relevant to do so to underline precisely how different they were.

We prefer not doing so for two reasons: 1) The information (as a band average) is already provided in Fig. 6, and 2) readability of Figs. 4-5 would be very low if we add two (three now, with the two Pinatubo case, see Reviewer #2 MC1) more spectra in each panel.

MaC5) Figure 7, although the uncertainties on the observations overlap with the result of the model presented, the two values are ~20% different. Instead of comparing the AE which is stated to be highly dependent on the pair of wavelengths selected to compute it, would it be possible to plot OMPS or SAGE-III channels on top of the extinction plot you give in fig. 5 to see if the slope is generally reproduce by the model with the assumptions used.

The AE depends on the pair of wavelengths chosen to calculate it but here we chose the same wavelengths for the observed SAGE/IASI and the modeled AE, so we think this is comparable and relevant. We added a statement to underline the 20% difference of observed versus modelled AE. On the contrary, the SAGE (and IASI) observations are not plottable in Fig. 4 (not Fig. 5) because this figure shows the extinction per unit {number density, mass}, which is not measured by satellite (there, the absolute extinction is measured).

**Minor comments:**

MiC1) Abstract: The abstract reads well, some explicit values (e.g. Angstrom Exp., mean radius) would increase the precision of the results in the abstract. In particular, AE computed between which wavelengths?

The wavelength interval for the AE calculations is now mentioned.

MiC2) Line 22: In two words, what was the main characteristic of Pinatubo and El Chichon? Larger aerosol sizes?

Except for the obvious much larger SO2, and then SA, injected mass, yes, the main difference of Pinatubo/El Chichon and Hunga eruptions are that the former have generally larger mean sizes and so significant modification of intensive optical properties of the volcanic aerosol layer, which translates to a different SW/LW radiative regime.

MiC3) Line 34-35: Please explicit the atypical conversion time to sulfates. Please, also add other references, maybe reflecting the range of results (e.g. Carn et al. 2022, Asher et al. 2023, …).
We added the information and cite now Carn et al. And Asher et al.

MiC4) Line 43: Limb observations or solar occultation? Duchamp et al. 2023 mainly used SAGE-III from what I read.
Yes, the Reviewer is totally right, we corrected this.

MiC5) Line 48: "A local maximum in number". Also see Norgren et al., 2024 (https://onlinelibrary.wiley.com/doi/abs/10.1029/2024JD040992).
Here we aimed at a very general (review) reference, which was chosen as Kremser et al., 2016. But Norgren et al., 2024 gives very interesting recent insights on the vertical distribution of UTLS aerosols, so we decided to add the reference here, even if without any further discussion.

MiC6) Table 2: Why not giving the AE for all the different wavelength pairs? One AE for Vis to MIR is trying to resume the complexity shown in figure 4 and 5 to a simple power law. It is shown to be wrong in this very study.
We understand the concern of the Reviewer #1. It is quite difficult to define an "average" AE, as the Reviewer also notices as a consequence of our results. This is true also when considering the different spectral intervals defined in Tab. 2, of course. If we would want to do that, how to chose the reference wavelength to compute the AE spectra to be averaged in each spectral interval? The underlying physics is well separated between two regimes: the scattering-dominated SW and the absorption-dominated LW. The first produces the power law and can be somewhat modelled, while the second depends strongly on the aerosol chemical composition and specific peaks on the imaginary part of the complex refractive index. One only AE – the one we provide – describes the power law occurring in the scattering-dominated SW but also the average extinction variability between these two regimes, and we think this is the best we can do looking at e.g. Fig. 4 and 6a.

**Reviewer #2**

This paper compares the optical impacts of stratospheric sulfate aerosol (SA) for background conditions and following Hunga-Tonga, Pinatubo, and El Chichón. The results are useful but the authors due a spotty job in the presentation and in characterizing the aerosol size distributions from Pinatubo and El Chichón. They also needlessly confuse things by introducing quantities that are not useful. These issues need to be addressed.
Thank you very much for your kind words and your constructive criticism. Please find a point-by-point reply to your major (MaC) and minor comments (MiC) in the following.

**Major Comments:**

MaC1) Following the eruptions of El Chichón there were very few measurements available to draw from, but that is not the case following Pinatubo when there were many measurements available both remote and in situ. So perhaps following El Chichón using just one reference to characterize the aerosol is perhaps justified. Still the effective radius determined of 1.7 μm is rather large. Perhaps it should be mentioned that this is 1.5 months after the eruption while 7 months later it was down to 0.9 μm (Hoffman and Rosen, 1983). But using just one measurement for Pinatubo is hardly justified, particularly when that reference Asano (1993) is misquoted. The distribution width from Asano is 1.05 not 1.5, which lowers the effective radius from 0.9 to 0.6 μm for

Pinatubo, which is likely more characteristic of other observations. Asano's estimate came within a month or so of the eruption. In any case the authors could do a much better job in finding an effective radius for Pinatubo. At least one that is more in line with most observations, or at least check that the observation they are using is consistent with other observations. The other major point that the authors ignore in the rush to characterize the effectiveness of the Hunga eruption is that the effective radius, and hence the concomitant optical effects, continuously evolve as the aerosol decays. For the comparisons here the time frame, e.g. the time since eruption, should be stated clearly.

First of all, thank you: we actually mistakenly used 1.5 instead of 1.05 for the SD width of Asano's paper! We corrected this and use the SD information and the effective radius (in effect 0.6 and not 0.9 μm) for the representation of a fresh Pinatubo plume stage, a period of time when the effect of the strong reduction of the SD width of the stratospheric aerosol layer, due to the Pinatubo eruption, is very evident, as discussed by Asano (1993). As shown by Russell et al. (1996), the SD of the Pinatubo plume evolved to larger effectivi radii, due to larger SD width, in the year following the event, and then the effective radius went slowly back to smaller background values. After 1 year since the eruption, different measurements shown by Russell et al. Showed extreme effective radii reaching values as large as 0.9 μm. Thus, we modelled an extreme case of a "1-year aged plume" as one with 0.9 μm effective radius. The two cases, young and 1-year aged plume, represent the variability range in the first-months-to-a-couple-years for the effective radius of the Pinatubo plume, this also responding the the Reviewer comment in terms of studying the optical properties of this plume during its evolution. We have modified all relevant tables and figures (Tab. 1, Fig. 6 and Fig. 8), as well as inherent text, including Abstract and Conclusions, to accunt for the extension of our comparsion to another SD for Pinatubo.

MaC2) 31 ... as high as 58 km ... 58 km is not "up to the lower mesosphere". What is this latter claim based on? It is not supported by Carr et al. 2020.

We corrected to "...at altitudes as large as 56 km...", which is the exact claim of Carr et al. 2022 (see also their Fig. 1a). This is normally in the lower mesosphere (the lower bound of the mesosphere is generally around 50 km, even if this can vary with latitude and season, of course), as also claimed by Carr et al. 2022 (see their Introduction). We corrected "Carr et al. 2020" to "Carr et al., 2022".

MaC3) 37-39 This sentence is superfluous and should be deleted. It is exactly what Zhu et al., 2022 showed.

We see this as a summary sentence that is necessary for the flow of the narrative of our Introduction, driving to the motivations of our study. We prefer to keep the sentence, even if we slightly reduced it in the revised text, but if the Reviewer #2 is still oriented in removing the sentence we might do in the next review round, in case.

MaC4) 39-41 This is not quite true. There were some CALIOPI observations of ash over one or two days as Sellitto et al. (2022a) state.

In effect our sentence was ambiguous and we modified it.

MaC5) 47 ... direct measurements with sondes balloons ... What kind of instrument is this?

This is decribed by Kloss et al., 2022 - see reference in the text. We clarify the sentence: "...with in situ balloon-borne optical counters measurements (Kloss et al., 2022)".

MaC6) 73-74 It should be mentioned that these are chosen as that was what was used by Hummel et al. In fact this point is mentioned later so this sentence can be deleted.

We rearranged the sentences to avoid redundancies.

MaC7) Fig. 2 The SDs shown are differential size distributions, i.e. dn/dr, so the units should be cm-3 µm-1 on the ordinate label. What number densities are used for the Pinatubo and El Chichón distributions? Are these still No=1 /cm3?
Corrected.

MaC8) 107-110 What is meant by "effective SA mass" and "effective number density"? How are they defined and where are the equations from? Why are they used?
See MaC9.

MaC9) What is the difference between effective SA mass and SA mass? Ultimately Me (Eq'n 2) is equivalent to the SA mass if one does the algebra. That is Me=4/3 pi ρ Re^3 Ne =4/3 pi ρ Re^3 No exp(-3 ln^2(σ)). Then Re=Rm exp(5/2 ln^2(σ)). Cubing Re and including it above leads to Me = 4/3 pi ρ Rm^3 exp(9/2 ln^2(σ)) which is exactly the SA mass for a unimodal lognormal distribution. So why all of these definitions rather than just say that: The SA mass can be recast as Eq'n 2 using Ne = No exp(-3 ln^2(σ))? Then ultimately since Me is just the SA mass why invent a new symbol rather than just call it the SA mass and define it as M? This needless additional definition of a different mass is unnecessary.
This is sometimes called "effective mass" just because it is calculated using "effective" number distribution and "effective" radius. But in fact, this is just mass (per unit volume) for a unimodal lognormal size distribution, as you say. We get how this can generate confusion and we mentioned this in the text.

MaC10) 115 Why are El Chichón and Pinatubo included in parentheses. Aren't they going to be playing a central role in this paper as points of comparison? Suggest … The optical properties of background, Hunga, El Chichón, and Pinatubo perturbed stratospheric aerosol layers are …
Agreed and done.

MaC11) Fig. 4 What are the parameters on the background distribution?
We are not sure to understand the question but, if the Reviewer #2 means "what are the yellow and red parameters noted in the caption over "Background"", these are the different possibilities for Hunga (colour-coded as in Fig. 2)

MaC12) 144 How is the number density fixed? In Table 1 it is fixed by setting the mass to a fixed value. Is that what is done here?
Yes, exactly.

MaC13) 146 Isn't this just the mass concentration?
Yes, what was defined as the "effective mass", see Ma8.

MaC14) 153 roto-vibrational? Is this rotational-vibrational or something else?
Corrected to rotational-vibrational.

MaC15) Figure 6 Panels b) and c) are mixed up.
Corrected

MaC16) Fig. 7 The information content in this figure is so low it doesn't justify a figure. The text is sufficient due to the small variation in the Angstrom exponent showing little difference between HT-HH and BG and both well within the uncertainty of the observations.
OK, we removed the figure and discuss this just in the text

MaC17) 238 What is TIR? FIR?

Yes, corrected

MaC18) 250—253 Split this four line long sentence into several sentences for readability.
We split this sentence in three.

MaC19) Fig. 8 Why make this figure difficult? Don't make the reader flip between positive and negative values, both shown on the same ordinate. Use an ordinate that reflects the quantities as they are; SW < 0, LW > 0, ... The LW is so small the axis would only have to run from -0.2 – +0.02. There is also a problem with the units given for the ordinate label.
We corrected the label. For the different RF signs, we just follow the usual way of showing this, as done e.g. in Lacis et al. We strongly prefer this way, which is actually much more readable.

MaC20) Considering that effective radii for various eruptions are a bit uncertain, and that they are always changing, as eruption sizes vary and the volcanic aerosol decays, why limit the calculations to just 4 effective radii? Using all the assumptions of Eq'n 4 the authors could produce a much more useful figure. Obviously there is structure in delta F, and it is not well reflected with just 4 calculations.
We definitely prefer showing just our cases, also considering that the RF dependence on the effective radius is not unique (RF depends on different optical properties of the aerosol layer, which depend on the full size distribution and composition). A denser sampling of the function would look like, to our eyes, a bit of an over-generalisation of this equation. Please also note that, due to the second Pinatubo case in our new manuiscript version, the figure is more "structured" than the previous version

MaC21) 257-260 This is not the first time this difference has been noted, e.g. (Li et al., 2024), as such it would be useful to compare these results with the previous work of others.
See Reviewer #1 MaC1.

MaC24) 265-266 Probably much smaller.
We agree.

MaC25) Table 2. There is plenty of room to use 7 columns in this table making it much easier to read and use. Column 1 is the same but can be reduced by 4 rows if it is followed by 6 more columns: BKG (Ae, SSA, g) and HT (Ae,SSA, g).
Yes, good idea.

MaC26) 294-295 Why is that? Is it because it is SA with not a large increase in effective radius, or?
Basically, yes

MaC27) 298-300 Doesn't this need to be qualified that the cooling rate is normalized by SA mass emitted? Then to get the total actual cooling one would have to multiply by the emitted mass.
This is clearly stated in the Results section.

**Minor Comments:**

MiC1) 76 Hummel's et al. or Hummel et al. The latter works fine.
Corrected

MiC2) 88 Not S but σ?
This is true: corrected

MiC3) 87 … In "this" work or "that" work? If the authors are implying Duchamp et al., then it should be "that". If they mean the paper here "this" is correct, but this latter doesn't seem to be the meaning.
We intended "in Duchamp et al." So we corrected to "that".

MiC4) 127 … has been calculated …
MiC5) 149 … in most of the …
Both corrected.

Mi6) 192 … were present …
It actually is "… a fraction… was present…"

Mi7) 291 …eruption has … or eruptions have …
Corrected to "had".

Li, C., Peng, Y., Asher, E., Baron, A. A., Todt, M., Thornberry, T. D., et al. (2024). Microphysical Simulation of the 2022 Hunga Volcano Eruption Using a Sectional Aerosol Model. *Geophysical Research Letters*, *51*(11), e2024GL108522. https://doi.org/10.1029/2024GL108522

---

## Author Response (AR2)

**Editor Comments**

Dear Authors,

I am pleased to share that the third reviewer is satisfied with your submission. Please consider addressing the reviewer comments. I am happy to accept your manuscript, pending mostly minor comments.

Thank you very much for the kind words and for the much appreciated serious work of supervision of the peer-review process for our paper. Please find our replies (in blue) to your comments (ECx) and those of the additional reviewer (RCx).

EC1) L10 "One notable effect of the Hunga eruption was the significant modification of the size distribution (SD) of the stratospheric aerosol layer with respect to background conditions and other recent moderate stratospheric eruptions, with larger mean particles size and smaller SD spread for Hunga." Please change to: In comparison to background conditions and other recent moderate stratospheric eruptions, one notable effect of the Hunga eruption was the significant modification of the size distribution (SD) of the stratospheric aerosol layer, resulting in larger mean particles size and smaller SD spread for Hunga.

Agreed and done

EC2) L34 Millàn -> Millán (i.e. please use an acute accent mark)

Sorry. Done.

EC3) L34 please add: Khaykin, et al https://doi.org/10.1038/s43247-022-00652-x; Vomel et al 10.1126/science.abq2299

Done

EC4) L40: with "The Hunga stratospheric aerosol perturbations were persistent at the southern-hemispheric scale" are you trying to say that it lasted (temporarily) longer than usual in the southern hemisphere or that it encompass the entire southern hemisphere? In any case persistent may not be the most suitable word. Please rephrase / clarify

We modified the sentence to "The Hunga stratospheric aerosol were quickly transported meridionally (weeks-to-months timescales) and stratospheric aerosol perturbations were soon observed encompassing the whole Southern Hemisphere (Legras et al., 2022, Taha et al., 2022). The Hunga aerosol layer showed optical signature of ash only for…"

EC5) L51: Missing period affer (Li et al., 2024).

Added, thanks.

EC6) L61 and the rest of the manuscript: El Chichon please change to: El Chichón

Yes, done in the text and figures

EC7) L63: I don't think this "This paper is structured as follows: in Sect. 2 the data 3 and methods used in the paper are described; in Section 3 results are shown and discussed; conclusions are drawn in Section 4. " is really needed. But you can leave it if you prefer.

We prefer keep it.

EC8) L71 and L78 Please use subscripts for H2SO4

Corrected.

EC9) L79: is 215K Really mid-stratospheric temperature. I think this temperature is really more like

lower stratospheric temperature. In any case, wasn't the aerosol plume mostly in the lower stratosphere?

Yes, but parts of the plume were at higher altitudes in the mid-stratosphere. We corrected the sentence to account for this.

EC10) L103 Consider deleting "in 1991" it was mentioned in the line above.
EC11) L109 Consider deleting "in 1982" it was already mentioned.
Yes, good idea. Done.

EC12) Figure 2: Please consider using thicker lines (also for Figure 4 and 5). Consider using the same x-axis limits in the panels so that the reader can easily intercompare the two. Why are the labels for sigma in red? The sigma distinction is made by the line type (i.e., dashed, solid, etc) so these legends should be gray (since it cannot be black because that color is being used for the background). Please change these sigma legends for figure 4 and 5 as well.
Thicker lines now done for Fig. 2. We tried the same for Figs. 4 and 5 but this made figures too busy, so we keep them as they were. Making x-axis limits the same in panels a and b makes actually panel a very less readable (red and yellow curves superposed) so we don't change but mention this in the caption. For the sigma colours: yes, very good idea! In our intentions, that red shade was kind of an "average" colours for the Hunga cases (orange to dark red) but we see that this is not so clear. So, we changed to grey (and same in Figs. 4 and 5, as well as in the Supplementary figure S1).

EC13) Figure 2 caption: There is no mention of the panel B. Although the description is there.
Added.

EC14) L139 Please consider rewriting this sentence "Values of the SSA approaching 1.0 (0.0) point at pure scattering (absorbing) particles" and L141 "Values of the g parameter approaching 1.0 (0.0) point at pure forward (isotropic) scattering." The use of this type of parenthetical structure can be confusing for many readers https://eos.org/opinions/parentheses-are-are-not-for-references-and-clarification-saving-space
Good point! Corrected.

EC15) L152 "and this latter impact is probably underestimated" Please consider changing to "likely leading to an underestimation of its impact."
Agreed and done

EC16) Figure 3 caption: "Scheme" change to "Schematic"
Done

EC17) Figure 4: The vertical dotted lines are barely visible. Also they are 2 different shades of blue ( I think by zooming a lot into the PDF). Please change the caption to something like: Light and dark blue and vertical lines indicate the Uv/Vis and MIR bands used in Figure 7.
The dotted lines are now thicker and we changed the figure caption.

EC18) Figure 6: Delete 1991
Done in the caption.

EC19) L276 ...SW scattering (pink triangle in Figure 7)
EC20) L277 ...the ageing of the plume, pink squares)
EC21) L278 ...SA, pink hexagon)
Yes, why not. Done.

EC22) Figure 7 caption: and Pinatubo (pink large squares) and El Chichón (pink hexagon)
Corrected

EC23) Table 2: I suggest adding one extra column stating Uv-Vis, NIR, MIR, and FIR
Yes, added

EC24) L 337 presumably missing comma in "(-0.012-0.040 and -0.015 …) Please double check
It actually was meanth to be an interval. We clartify in the text now

**Reviewer #3 Comments**

The primary outcome of this study is that the authors used size distributions provided by Duchamp et al. (2023), for both the Hunga perturbed and the background stratospheric aerosol layer to determine aerosol optical properties. Then the aerosol properties has been compared with other volcanic eruptions. This work used the Mie code provided by my current group (EODG) in Oxford. The manuscript is written well, and the results are discussed in an appropriate way. I would like to recommend this manuscript for publication with minor corrections. The authors have already put in significant effort, especially as the manuscript has undergone a round of review by other reviewers. I do not wish to add any extra burden; however, I hope the minor comments listed below will be helpful if the authors choose to consider these changes. Overall, the manuscript is well-structured, easy to follow, and a pleasure to read.
Thank you for your kind words. Please find our replies to your comments in the following.

RC1) Line 42: "The Hunga stratospheric aerosol perturbation can then solely be associated to SA".
This might be mostly due to the look-up tables. These LUTs might be missing the crucial information about the stratospheric ash particles. Thus it would be very nice to mention this in this line.
It actually is an evidence of many previous studies, including different satellite, ground-based and in situ measurements, including Sellitto et al., 2022a and Legras et al., 2022 cited in the Introduction of the present manuscript.

RC2) Line 65-66: "As discussed in the introduction, both the background stratospheric aerosol layer and its perturbation brought by the Hunga eruption can be solely characterised, in terms of composition, as secondary SA particles." This sentence is not simply true, as I mentioned that the satellite based retrieval algorithms might be missing the crucial information about the presence of Ash particles. Please mention this in this sentence. Thus the assumption of SA are only considered. This logic makes sense.
See my reply to RC1.

RC3) Line 76: For the selection of 215K temperature, is it possible cite some recent publications instead of 1988 publication? Because due to climate change the stratosphere is rapidly warming.
Hummel et al. (1988) do not discuss stratospheric temperatures and their trends but just describes his CRI dataset, from laboratory measurements, at different temperatures. This dataset have SA CRIs at just a limited number of temperatures, but is the one we can use because of the simultaneous availability of SW and LW CRIs (which is not often the case in CRI databases). This CRI temperature is the most adapted, among those available, to represent the lower and mid-stratospheric conditions (see also EC9).

RC4) Further, is it possible to increase the figure quality?
Not in terms of the spectral resolution because this is the full spectral resolution of the Hummel et al. (1988) CRI dataset, unfortunately.

RC5) Would it be possible to add a paragraph in the conclusion section discussing future directions? Specifically, it would be valuable to outline potential approaches for improving results and comparisons when analyzing the Hunga eruption, which has more extensive observational data, in contrast to the El Chichón and Pinatubo eruptions, where such data is limited.

We added a sentence on the importance of aerosol SD for radiative impact estimations.